# A self-supervised domain-general learning framework for human ventral stream representation

Talia Konkle [1✉] & George A. Alvarez[1✉]

Anterior regions of the ventral visual stream encode substantial information about object categories. Are top-down category-level forces critical for arriving at this representation, or can this representation be formed purely through domain-general learning of natural image structure? Here we present a fully self-supervised model which learns to represent individual images, rather than categories, such that views of the same image are embedded nearby in a low-dimensional feature space, distinctly from other recently encountered views. We find that category information implicitly emerges in the local similarity structure of this feature space. Further, these models learn hierarchical features which capture the structure of brain responses across the human ventral visual stream, on par with category-supervised models. These results provide computational support for a domain-general framework guiding the formation of visual representation, where the proximate goal is not explicitly about category information, but is instead to learn unique, compressed descriptions of the visual world.

[1] Department of Psychology & Center for Brain Science, Harvard University, Cambridge, MA, USA. ✉email: talia_konkle@harvard.edu; alvarez@wjh.harvard.edu

Patterned light hitting the retina is transformed through a hierarchy of processing stages in the ventral visual stream, driving to a representational format that enables us to discriminate, identify, categorize, and remember thousands of different objects[1–6]. What pressures govern the formation of this visual representation? This is a deeply debated question, with proposals balancing the relative guiding influence of innate biases versus the structure of the visual input statistics, and the degree to which learning operates over more domain-specialized versus domain-general architectures[3,7–13].

For example, some prominent theoretical accounts of high-level visual system organization assert that the representations are explicitly about object categories, and that category-level ("domain-level") forces are critical for guiding this visual representational format[7,11,14–16]. For example, these theories argue that visual representation formation is guided by distinct long-range network connectivity to help learn features in support of broad conceptual distinctions (e.g. for animate or inanimate entities[14,15]), or to guide the learning of a specific set of categories with particular functional relevance (e.g., faces, bodies, and scenes[3,11]).

And, in what has sometimes been taken as converging support for the role of category-level pressures in forming visual representations, deep convolutional neural network models—trained directly to support object categorization—develop hierarchical feature spaces that show an emergent match with brain responses[17–24] (see refs. [25,26] for review). However, on deeper examination, it is not clear that the category-level supervisory signals involved in training deep neural networks are a good proxy for the representational pressures implied in the domain-level cognitive neuroscience theories. In particular, these deep neural networks models are trained with much finer-grained distinctions at the subordinate category level (e.g., with features that are explicitly guided to differentiate among three different kinds of crabs, not to mention the 90 breeds of dogs, present among the 1000 categories of the ImageNet database[27]). Thus, it is clear these category-supervised deepnet models are operationalizing category-level pressures in a different way than is generally posited by most cognitive neuroscience theories.

Alternate theories of visual representation formation put relatively more weight on the structure of the natural image statistics, and relatively less weight on downstream output needs driving visual representation formation[9,10,12,28,29]. These theoretical proposals argue that the visual cortex is a generic covariance extractor and that there are systematic differences in the way things look (e.g., among faces and scenes; among animals, big objects, and small objects)–it is these perceptual feature differences that actually underlie the 'categorical' distinctions of high-level visual responses[28,30,31]. On this account, visual learning is less a process of enrichment (i.e., building new features for each new category) and more a process of differentiation (i.e., learning to seek out distinguishing features that are already present in the visual input[32,33]). A strong version of this domain-general theoretical framework posits that learning good visual representation does not at all rely on presupposing categories, leveraging labels, or otherwise drawing on any specialized architectural constraints for some kinds of stimuli over others. However, a key challenge remains to make this domain-general proposal more computationally explicit: what is an alternative representation goal, if not category-supervision, that might serve as an internal learning signal to draw out useful structure from natural image statistics?

Key insight into this challenge comes from work that changed the objective from learning features that can discriminate all categories from each other to instead learning features that can discriminate every view from every other view[34]. The logic here is that that views of objects from the same category will naturally project nearby in such a feature space, due solely to the statistical structure of the input and the generic architectural prior, without explicit category-level pressure. Inspired by this insight, here we developed a learning framework that is fully self-supervised, called instance-prototype contrastive learning (IPCL). The model operates by taking multiple samples over an image and projecting these through a deep convolutional neural network backbone into a low-dimensional embedding space. To learn instance-level structure, the model tries to map these multiple samples of the views nearby by in the latent space (towards the "instance-prototype"), while also making this embedding distinct from the representations of recently encountered views ("contrastive learning"). As such, the final representational format can be conceived of as a high-fidelity perceptual representation, capable of fine-grained discrimination between views. Within the broader cognitive neuroscience context, this model thus operationalizes a domain-general view of visual representation learning, where no specialized pressures beyond the visual system are required to guide the format of visual representation.

In the present work, we show that our instance-prototype contrastive learning models indeed have naturally emergent category structure in the latent space. Further, these models learn hierarchical visual feature spaces that can capture brain response structure, on par with category-supervised models, at or near the noise ceiling of the data in most regions, in two condition-rich fMRI datasets. Concurrent with the present work, and at an extremely rapid pace, new self-supervised instance-level contrastive learning models have been introduced which have even higher emergent categorization accuracy[35–40]; however, we find the representational spaces learned in these more performant feature spaces are not increasingly more brain-like (c.f. ref. [23]). As a whole, this work invites a shift away from the category-based specialized framework that has been dominant in high-level visual cognitive neuroscience, providing an alternative conceptual framework in which the representational goal of the visual system is to capture fine-grained visual differences in a useful compressed format, learnable with domain-general mechanisms. Critically, for this argument, the models are not intended to be direct models of the biological brain per se, but rather to serve as computational existence proofs of what kind of representational formats are learnable from the input given certain constraints. As such, the degree to which these models learn representational formats that show correspondence with the visual system provides computational-empirical plausibility for a domain-general view of the formation of visual system representation.

## Results

**Instance-prototype contrastive learning**. We designed an instance-prototype contrastive learning algorithm (IPCL) to learn a representation of visual object information in a fully self-supervised manner, depicted in Fig. 1a. The overarching goal is to learn a low-dimensional embedding of natural images, in which sampled views of the same image are nearby to each other in this space and also separable from the embeddings of all other images.

To do so, each image is sampled with 5 augmentations, allowing for crops, rescaling, and color jitter (following the same parameters as in ref. [41]). These samples are passed through a deep convolutional neural network backbone and projected into a 128-dimensional embedding space, which is L2-normed so that all image embeddings lie on the unit hypersphere. The contrastive learning objective has two component terms. First, the model tries to make the embeddings of these augmented views similar to each other by moving them towards the average representation among these views—the "instance prototype." Simultaneously, the model tries to make these representations dissimilar from those of recently encountered items, which are stored in a

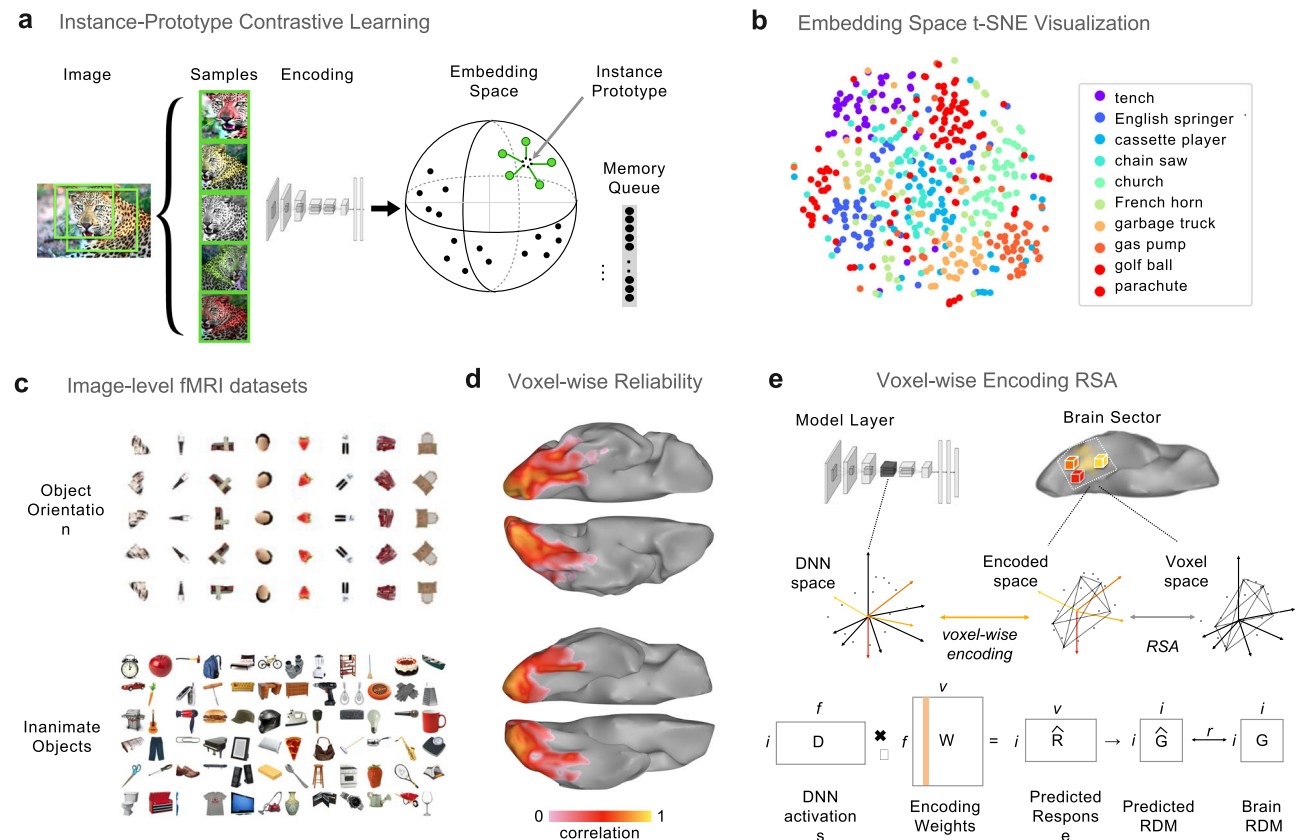

**a** Instance-Prototype Contrastive Learning

**b** Embedding Space t-SNE Visualization

- tench
- English springer
- cassette player
- chain saw
- church
- French horn
- garbage truck
- gas pump
- golf ball
- parachute

**c** Image-level fMRI datasets

**d** Voxel-wise Reliability

**e** Voxel-wise Encoding RSA

0 ——— 1
correlation

**Fig. 1 Model overview and methodological approach. a** Overview of the self-supervised instance-prototype contrastive learning (IPCL) model which learns instance-level representations without category or instance labels. **b** t-SNE visualization of 500 images from ten ImageNet categories, showing emergent category clusters in deepnet feature space. **c** All stimuli for the two fMRI datasets. Note that in this figure, the face image has been covered, to remove identifying information. **d** View from the bottom of the brain, showing voxel-wise reliability across the ventral visual stream for the Object Orientation dataset (top) and Inanimate Objects dataset (bottom). The color bar indicates the Pearson correlation between odd and even halves of the data. **e** Overview of the voxel-wise encoding RSA procedure. Source data are provided as a Source Data file.

lightweight memory queue of the most recent 4096 images—the "contrastive" component. See the Supplementary Information for the more precise mathematical formulation of this loss function.

For the convolutional neural network backbone, we used an AlexNet architecture[42], modified to have group-normalization layers ($N = 32$ groups) rather than standard batch normalization; see Supplementary Fig. 1), which was important to stabilize the learning process. While traditional batchnorm normalizes each individual channel across the full image batch, groupnorm normalizes across groups of channels for each individual image[43], group normalization operates by normalizing across groups of feature channels for each image[34], with intriguing parallels to divisive normalization operations in the visual system[44,45]. Five IPCL models were trained under this learning scheme, with slightly different training variations; all training details can be found in the Supplementary Information.

**Emergent object category information.** To examine whether these self-supervised models show any emergent object category similarity structure in the embedding space, we used two standard methods to assess 1000-way classification accuracy on ImageNet. The k-nearest neighbor (kNN) method assigns each image a label by finding the k (=200) nearest neighbors in the feature space, assigning each of the 1000 possible labels a weight based on their prevalence amongst the neighbors (scaled by similarity to the target), and scoring classification as correct when the top-weighted class matched the correct class (top-1 knn accuracy[41]). The linear evaluation protocol trains a new 1000-way classifica-

tion layer over the features of the penultimate layer to estimate how often the top predicted label matches the actual label of each image[38,39] (see Supplementary Information for method details).

Object category read-out from the primary IPCL models achieved an average top-1 kNN accuracy of 37.3% (35.4−38.4%) from the embedding space and 37.1% (32.2−39.7%) from the penultimate layer (fc7). In contrast, untrained models with a matched architecture show minimal object categorization capacity, with top-1 kNN accuracy of 3.5% (3.3−3.8%) and top-1 linear evaluation accuracy of 7.2% (fc7). Figure 1b visualizes the category structure of an IPCL model, showing a t-SNE plot with a random selection of 500 images from ten categories, arranged so that images with similar IPCL activations in the final output layer are nearby in the plot. It is clear that images from the same category cluster together. Thus, these fully self-supervised IPCL models have learned a feature space, which implicitly captures some object category structure, with no explicit representational pressure to do so.

For comparison, we trained a category-supervised model with matched architecture and visual diet and tested the categorization accuracy with the same metrics as the self-supervised model. The kNN top-1 accuracy was 58.8%, with a linear readout of 55.7% from the penultimate layer (fc7). An additional category-supervised matched-architecture model, trained with only one augmentation per image (rather than 5, which is a more standard training protocol), also showed similar classification accuracy (readout from fc7: kNN top-1: 55.5%; linear evaluation top-1: 54.5%). Thus, these matched-architecture category-supervised

models have notably better categorization accuracy on the ImageNet database than our IPCL-trained models. Supplementary Table 1 reports the categorization accuracies for all of the individual models.

**Relationship to the structure of human brain responses**. To the extent that categorization capacity is indicative of brain-like representation in this accuracy regime[23], we would expect these fully self-supervised models to have feature spaces with at least some emergent brain-like correspondence, but not as strong as category-supervised models. However, it is also possible that feature spaces learned in these self-supervised models have comparable or even more brain-like feature spaces than category-supervised models (e.g., if the instance-level representational goal more closely aligns with that driving visual system tuning). Thus, we next examined the degree to which the IPCL feature spaces have an emergent brain-like correspondence, relative to the category-supervised models.

Brain responses were measured using functional magnetic resonance imaging (fMRI) in two different condition-rich experiments (Fig. 1c, see Methods and Supporting Information), using a powerful 4 s mini-block design that provides reliable estimates of responses to individual items[46]. The Object Orientation dataset included images of eight items presented at five different in-plane orientations; this stimulus set probes for item-level orientation-tolerance along the ventral visual hierarchy, while spanning the animate/inanimate domain. The Inanimate Objects dataset included images of 72 everyday objects; this stimulus set probes finer-grained distinctions within the inanimate domain. Thus, these two stimulus sets provide complementary views into object similarity structure. The resulting data revealed reliable voxel-level responses along the ventral visual stream (Fig. 1d; see Methods). To delineate brain regions along the hierarchical axis of the ventral stream, we defined three brain sectors reflecting the early visual areas (V1–V3), the posterior occipito-temporal cortex (pOTC), and the anterior occipito-temporal cortex (aOTC; see Methods). Within these sectors, individual subject representational geometries were reliable and consistent across subjects, yielding highly reliable group-averaged representational geometries (EarlyV split-half reliability: $r = 0.86$–$0.90$; pOTC: $r = 0.75$–$0.90$; aOTC: $r = 0.60$–$0.89$), providing a robust target to predict with different deep neural networks.

To relate the representations learned by these deep neural networks with brain sector responses along the ventral visual hierarchy, we used an approach that leveraged both voxel-wise encoding methods[47,48] and representational similarity[49], which we subsequently refer to as voxel-wise-encoding RSA (veRSA; Fig. 1e; see Methods; see also refs. [50,51]). This method fits an encoding model at each voxel independently, using weighted combinations of deepnet units ($W$), to predict the univariate response profile. Then, the set of voxel encoding models are used to predict multi-voxel pattern responses to new items ($\hat{R}$) and to derive the predicted representational geometry in this encoded space ($\hat{G}$). This predicted RDM is then compared to the RDM of the brain sector ($G$), as the key measure of how well the layer's features fit that brain region. This analysis choice places theoretical value on the response magnitude of a voxel as an informative brain signature, while also reflecting the theoretical position in which neurons across the cortex participate as a unified population code.

The brain predictivity of the models are depicted in Fig. 2. The results show that the IPCL model achieves parity with the category-supervised models in accounting for the structure of brain responses, evident across both datasets and at all three levels of hierarchy. Each plot shows the layer-wise correlations between the predicted and measured brain representational geometry,

with all IPCL models in blue (with multiple lines reflecting replicates of the same model with slight training variations, see Methods), and category-supervised models in orange. The adjacent plots show the maximum model correlation, reflecting the layer with the strongest correlation with the brain RDM, computed with a cross-validated procedure to prevent double-dipping (cv max-r; see Methods), plotted for IPCL models, category-supervised models, and an untrained model. Supplementary Table 2 reports the statistical tests comparing the brain predictivity between IPCL and category-supervised models, e.g., in 56/60 comparisons, the cross-validated max correlation for the IPCL models is greater than or not significantly different from category-supervised models (and with Bonferroni correction for multiple comparisons category-supervised models never showed a significantly higher correlation than an IPCL model).

Further, all models account for a large proportion of the explainable variance in these highly- reliable brain representational geometries—though with a noticeable difference between the two datasets. Considering the object orientation dataset, the proportion of explainable variance accounted for approached the noise ceiling in all sectors for both IPCL and the category-supervised models (mean IPCL: 88, 84, 94; category-supervised: 82, 91, 87; noise ceiling: $r = 0.90, 0.90, 0.89$; for EarlyV, pOTC, and aOTC, respectively). However, considering the Inanimate Objects dataset, neither the IPCL nor category-supervised counterpart models learned feature spaces that reached as close to the noise ceiling, leaving increasing unaccounted for variance along the hierarchy (mean IPCL: 74, 47, 32%; category-supervised: 65, 41, 28%; noise ceiling: $r = 0.86, 0.74, 0.60$; for EarlyV, pOTC, aOTC, respectively). These results reveal that the particular stimulus distinctions emphasized in the dataset matter, as these dramatically impact the claim of whether the representations learned by these models are fully brain-like, or whether the models fall short of the noise ceiling.

Finally, these results also generally show a hierarchical convergence between brains and deep neural networks, with earlier layers capturing the structure best in the early visual cortex, and later layers capturing the structure in the occipito-temporal cortex. Unexpectedly, we also found that the untrained models were competitive with the trained models in accounting for responses in EarlyV and partially in pOTC, whereas both IPCL and category-supervised models clearly outperform untrained models in aOTC. A deeper inspection revealed that the predicted representational distances in untrained models hover around zero, which is consistent with the fact that they cannot classify object categories very well. However, these feature spaces nevertheless contain small differences that are consistent with the brain data, amplified by the voxel-wise encoding procedure. Further, the use of Group-Normalization layers also boost untrained models—e.g., local Response Norm or Batch Normalization generally fit brain responses less well, particularly in the early visual cortex (see Supplementary Fig. 2). These findings highlight that there are useful architectural inductive biases present in untrained networks.

Overall, these results show that our instance-prototype contrastive learning models, trained without category-level labels, can capture the structure of human brain responses to objects along the visual hierarchy, on par with the category-supervised models. This pattern holds even in later stages of the ventral visual stream, where inductive biases alone are not sufficient to predict brain responses.

**Varying the visual diet**. As some of the reliable brain responses in the later hierarchical stages of the Inanimate Objects dataset was unexplained, we next explored whether variations in the

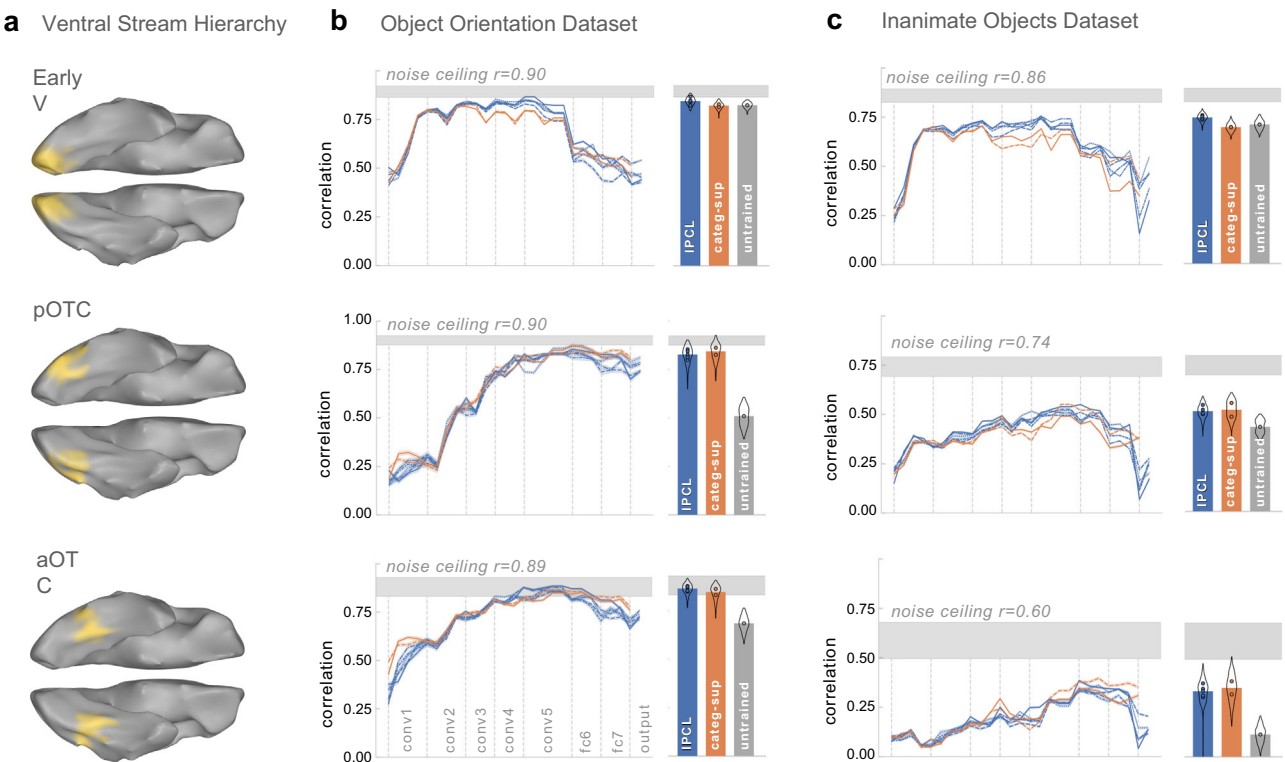

**Fig. 2 Model-to-brain fits. a** Visualization of the ventral stream regions of interest spanning the visual hierarchy from posterior to anterior (EarlyV, pOTC, aOTC). **b** and **c** show the veRSA results for the Object Orientation and Inanimate Object datasets, respectively. Each panel plots the mean correlation between model RDMs with neural RDMs (y-axis), averaged over split-halves of the brain data, shown separately for each model layer (x-axis) and brain region (rows). All IPCL models are in blue, and category-supervised models are in orange. The thickness of each line reflects 95% confidence intervals based on 1000 bootstrapped samples across split-halves. Bar plots show cross-validated estimates of the maximum correlation across model layers for each model class (IPCL in blue, category-supervised in orange, and an untrained model in gray). Error bars reflect a mirrored density plot (violin plot) showing the distribution of correlations across all split-halves, aggregated across instances of a given model type. Distributions are cutoff at ±1.5 IQR (interquartile range, Q3-Q1). Source data are provided as a Source Data file.

visual diet of the IPCL models might increase their brain predictivity. For example, the pressure to learn instance-level representations over a more diverse diet of visual input might result in richer feature representations that better capture the structure neural representations, particularly in the later brain stages reflecting finer-grained inanimate object distinctions. However, it is also possible that the relatively close-scale and centered views of objects present in the ImageNet database are critical for learning object-relevant feature spaces, and that introducing additional content (e.g., from faces and scenes) will detrimentally affect the capacity of the learned feature space to account for these object-focused brain datasets.

To probe the influence of visual diet, we trained six new IPCL models over different training image sets (Fig. 3a; see Methods, Supplementary Information, and Supplementary Table 1), and compared their brain predictivity to the ImageNet trained baseline. First, because we made some changes to the image augmentations to accommodate all image sets, we trained a new baseline IPCL model on ImageNet. Second, we used object-focused images from a different dataset as a test of near-transfer (OpenImages[52,53]). The third dataset was scene images (Places2, ref.[54], which we consider an intermediate-transfer test, as models trained to do scene categorization also learn object-selective features[55]. The fourth dataset was faces (VGGFace2; ref.[56], a far-transfer test that allows to explore whether a visual diet composed purely of close-up faces learns features that are sufficient to capture the structure of brain responses to isolated objects. The fifth dataset included a mixture of objects, faces, and places,

which provides a richer diet that spans traditional visual domains, with the total number of images per epoch matched to the ImageNet dataset. The sixth dataset had the same mixture but used three times as many images per epoch to test whether increased exposure was necessary to learn useful representations with this more diverse dataset.

For each of these six models trained with different kinds of visual experience, we used the same veRSA approach and then calculated the cross-validated maximum correlation across layers (see Methods). The results are plotted in Fig. 3b, where the five IPCL models with different visual experiences (colored violin plots) are plotted in the context of the new baseline IPCL model trained on ImageNet (black dashed lines).

The overarching pattern of results shows that the visual diet actually had very little effect on how well the learned feature spaces could capture the object similarity structure measured in the brain responses. Quantitatively, the mean absolute difference in brain predictivity from the baseline ImageNet- trained model was $\Delta r < 0.044$ (range of signed differences −0.202 to 0.040). The visible outlier is the model trained only with views of faces. The features learned by this model were significantly less able to capture the structure of the object orientation dataset in both the posterior and anterior occipito-temporal cortex, with a difference from the baseline model >2.5 standard deviations from the mean difference across all comparisons (pOTC: $z = 3.67$; aOTC: $z = 3.21$). However, the feature spaces of this model were still able to capture the differences among objects in the Inanimate Object dataset, variants in EarlyV and pOTC (though with a

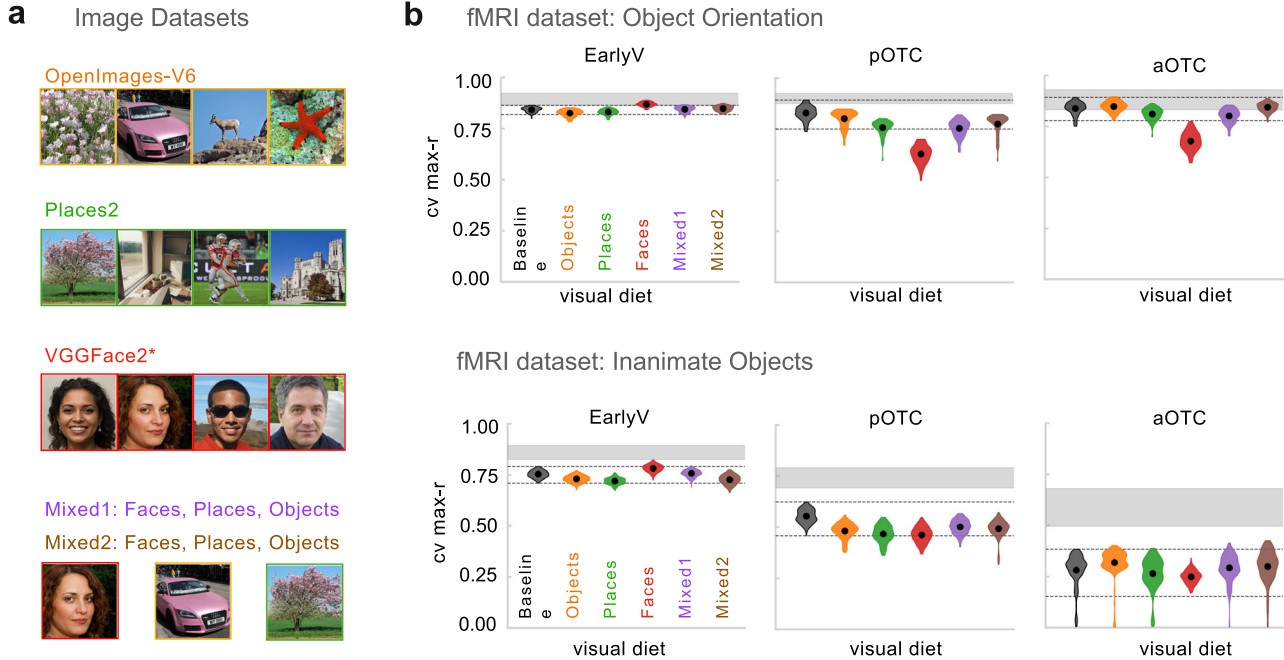

**Fig. 3 Consequences of variation in the visual diet. a** Example images in the style of each of dataset are shown. For OpenImagesV6 and Places2, similar images were found from commons.wikimedia.org. For VGGFace2, images were generated from thispersondoesnotexist.com. **b** The cross-validated maximum correlation (cv max-r) between model RDMs and neural RDMs for each dataset (rows), and each brain region (columns). Mean scores are shown with a black dot at the center of a mirrored density plot (violin plot) showing the distribution of correlations across all split-halves (distributions are cutoff at ±1.5 IQR, interquartile range, Q3-Q1). The dashed black lines indicate the ±1.5 IQR range for the matched baseline IPCL model trained on ImageNet. Source data are provided as a Source Data file.

small reliable difference in pOTC) and was not different from the ImageNet trained baseline in aOTC (corrected t < 1). The full set of results are reported in Supplementary Table 3.

Overall, this second set of IPCL models suggests that the statistics of most natural input contains the relevant relationships to comparably capture these brain signatures. Further, these models also highlight the general nature of the learning objective, demonstrating that it can be applied over richer and more variable image content, which is traditionally learned separately in supervised learning.

**Accuracy vs brain predictivity**. The analyses so far demonstrate that, while category-supervised models show better object categorization capacity, IPCL models still achieve parity in their correspondence with the visual hierarchy. However, neither the category-supervised nor the IPCL models are able to fully capture the structure of the measured brain responses, particularly in the later hierarchical stage of the Inanimate Objects dataset that captures many finer-grained object relationships. This predictivity gap raises a new question— if instance-level contrastive learning systems advance to the point of achieving comparable emergent classification accuracy to category-supervised models, will even more brain-like representation emerge?

Concurrently, a number of new instance-level contrastive learning models have been developed, which allow us to test this possibility (e.g., SimCLR[39], MoCo[37], MoCoV2[38], and SwAV[40]). For example, SimCLR leverages related principles as our IPCL network, with a few notable differences: it uses two augmentations per image (rather than an instance prototype), a more compute-intensive system for storing negative samples (in contrast to our lightweight memory queue), and a more powerful architectural backbone (Resnet50; ref. [57]). Critically, this model, and others like MoCoV2 and SwAV, now achieve object classification performance that rivals their category-supervised comparands. Do these models show more

brain-like representation, specifically in their responses to inanimate objects, where the later hierarchical brain structure was reliable and unaccounted for?

The results indicate that these newer models do not close this gap. Figure 4 depicts the relationship between top-1 accuracy and the strength of the brain correspondence, for the Inanimate Object dataset. All instance-level contrastive learning models are plotted with colored markers, while category-supervised models are plotted with open markers. Different base architectures are indicated by the marker shape). These scatter plots highlight that, across these models, top-1 accuracy ranges from 26–73%; however, improved categorization capacity is not accompanied by a more brain-like feature space. Further, these plots suggest that these particular variations in architecture, including higher powered ResNet[57] and ResNeXt[58] models, also do not seem to close this gap.

Finally, we also asked whether a recent self-supervised model trained on an, even more, ecological visual diet—images sampled from baby head-mounted cameras—might show better brain predictivity (TC-Moco[59], SAYCam dataset[60]). The visual experience of toddlers involves extensive experience with very few things, rather than an equal distribution over many categories–a visual curriculum which may be important for visual representation learning[61]. However, this particular model also did not close the brain predictivity gap evident in the similarity structure of inanimate objects at the later stages of the visual hierarchy (Fig. 4; purple diamond). Note though that this model does not yet take advantage of temporal information in videos beyond a few frames; building effective systems that use contrastive learning over video is an active frontier[62–64].

Overall, the Inanimate Objects dataset has revealed some reliable representational structure in the object-selective cortex that is not easily captured by current deepnet models, even across these broadly sampled variations in learning algorithm, architecture, and visual diet. Further, these aggregated results

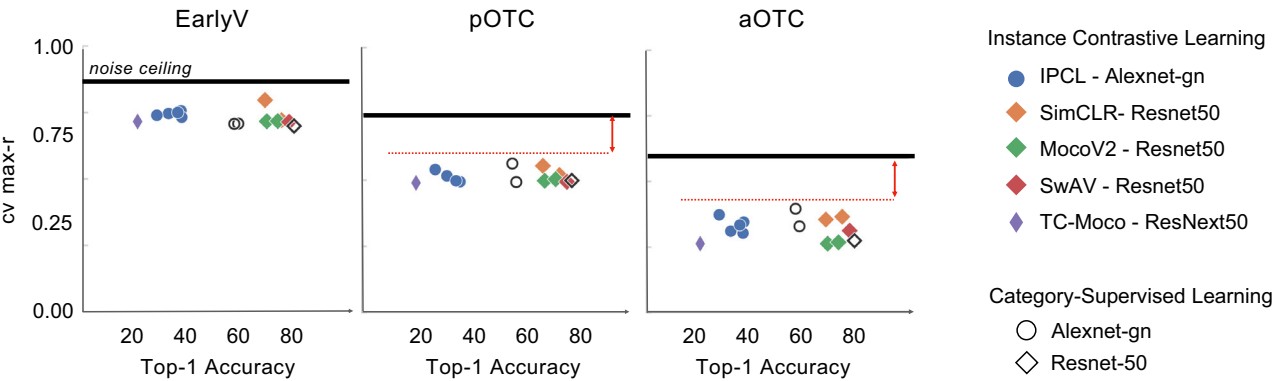

**Fig. 4 Relationship between object classification accuracy and brain predictivity.** The x- axis plots top-1 classification accuracy, and the y-axis plots the cross-validated max correlation with the Inanimate Object dataset, in each of the three brain sectors. Self-supervised contrastive learning models are shown with colored markers and category-supervised with open markers. Model architecture is indicated by marker shape. Red dashed lines and double-headed arrows draw attention to the gap between these model fits and the reliability ceiling of these brain data. Source data are provided as a Source Data file.

complement the emerging trend that overall object categorization accuracy is not indicative of overall brain predictivity[23], here considering a variety of other instance-level contrastive learning methods, over a much wider range of top-1 accuracy levels.

**Auxiliary results**. For reference, we also conducted the same analyses using a classic representational similarity analysis (rather than veRSA), in which there was no voxel-wise encoding models, nor any deepnet unit feature re-weighting (Supplementary Figs. 2–4). Overall, the magnitude of the correlation between model layers and the brain RDMs was systematically lower than when using veRSA. Despite this general main effect, the primary claims were also evident in this simpler analysis method: IPCL models showed parity with (or even superior performance to) category-supervised models, across brain sectors and datasets, with one notable exception. That is, in the aOTC and when considering the object orientation dataset, the category-supervised model showed better correspondence with the brain than the IPCL models (Supplementary Fig. 3). This discrepancy between classic RSA and veRSA does highlight that veRSA is able effectively to adjust the representational space to better capture the brain data, while classic RSA weights all features equally. We discuss these results in the context of the open challenge of linking hypotheses between deepnet features and brain responses.

## Discussion

Here we introduced instance-prototype contrastive learning models, trained with no labels of any kind, which learn a hierarchy of visual feature spaces that (i) have emergent categorization capacity based on the local similarity structure of the latent space and (ii) predict the representational geometry of hierarchical ventral visual stream processing in the human brain, on par with category-supervised counterparts. This correspondence with the structure of human visual system responses held in two datasets, considering both object orientation variation and finer-grained inanimate object distinctions, and also held over self-supervised models trained with different visual input diets. Finally, we highlight that there is a representational structure in the brain that was not well accounted for by any model tested, particularly in the anterior region of the ventral visual stream, related to finer-grained differences among inanimate objects. Broadly, these results provide computational plausibility for instance-level separability—that is, to tell apart every view from

every other view–as a plausible goal of ventral visual stream representation, which reflects a shift away from the category-based framework that has been dominant in high-level visual cognitive neuroscience research.

**Implications for the biological visual system**. The primary advance of this work for insights into the visual system is to make a computationally supported learnability argument: it is possible to achieve some category-level similarity structure without pre-supposing explicit category-level pressures. Items with similar visual features are likely to be from similar categories, and we show that the goal of instance-level representation allows that natural covariance of the data to emerge in the latent space of the model — a result that is further supported by the expanding set of self-supervised models with emergent object categorization accuracy comparable to category-supervised systems[36–40]. Our work adds further support for the viability of this hypothesis of visual system representation, by demonstrating an emergent correspondence with the similarity structure measured from brain responses—e.g., it is not the case that our self-supervised models learn a representation format that is decidedly un-brain-like. Indeed, recent work suggests that not all self-supervised learning objectives achieve brain-like representation with parity to category-supervised models[65].

Our model invites an interpretation of the visual system as a very domain-general learning function[10], which maps undifferentiated, unlabeled input into a useful representational format. On this view, the embedding space can be thought of as a high-fidelity perceptual interface, with useful visual primitives over which separate conceptual representational systems can operate. For example, explicit object category-level information may be the purview of more discrete compositional representational systems, that can provide "conceptual hooks" into different parts of the embedding space[66–68]. Intriguingly, new theoretical work suggests that instance-level contrastive learning may actually implicitly be learning to invert the generative process (i.e., mapping from pixels to the latent dimensions of the environment which give rise to the projected images[69], suggesting that contrastive learning may be particularly well-suited for extracting meaningful representations from images.

What does the failure of these models to predict reliable variance in aOTC for the Inanimate Objects dataset tell us about the nature of representations in this region? Using this same brain

dataset, we have found that behavioral judgments related to the shape similarity, rather than semantic similarity, show better correspondence with aOTC[70–72]. This result raises the possibility that the deepnets tested here are missing aspects of shape reflected in aOTC responses (e.g. structural representations[73], global form[74], or configural representations[75]), which resonates with the fact that deep convolutional neural networks (CNNs) operate more as local texture analyzers[76,77], and may be architecturally unable explicitly represent global shape[78]. Taken together, these results indicate that the success of CNNs in predicting ventral stream responses is driven by their ability to capture texture-based representations that are also extensively present throughout the ventral stream[28], but they fall short where more explicit shape representations are emphasized. Capturing brain-like finer-grained distinctions among inanimate objects is thus an important frontier that is currently beyond the scope of both contrastive and category-supervised CNN models.

**Components of the learning objective**. Why is instance-prototype contrastive learning so effective in forming useful visual representations, and what insights might this provide with respect to biological mechanisms of information processing? Recent theoretical work[79] has revealed that the two components of the contrastive objective function have two distinct and important representational consequences, which they refer to as alignment (similarity across views) and uniformity (using all parts of the feature space equally). To satisfy the alignment requirement, the model must learn what it means for images to be similar. For IPCL, the model takes five samples from the world and tries to move them to a common part of the embedding space, forcing the model to learn that the perceptual features shared across these augmentations are important to preserve identity, while the unshared perceptual features can be discarded. Interpreted with a biological lens, these augmentations are like proto-eye movements, and this analogy highlights how this model can integrate more active sensing and predictive coding[80]. For example, augmentations could sample over translation and rotation shifts of the kind that occur with eye and head movements. Further, "efference copy" signals[81,82], which signal the magnitude and direction of movements between samples, might also lead to predictable shifts in the embedding space. This intrinsic information about the sampling process could enable the system to learn representations that are "equivariant", as opposed to "invariant", over identity-preserving transformations[83,84].

The second component of the objective function enforces representational uniformity–that is, where the set of all images have uniform coverage over the hypersphere embedding space. In IPCL this is accomplished by storing a modest set of "recent views" in a memory queue to serve as negative samples; other successful contrastive learning models use a much larger set of negatives (either in a batch or queue) which presumably helps enforce this goal[38,39]. The memory queue also has biological undertones: the human and non-human primate ventral streams are effectively a highway to the hippocampus[85]. Through this lens, the recent memory queue of IPCL is a stand-in for the traces that would be accessible in a hippocampal memory system, inviting further modifications that vary the weight of the contrast with fading negative samples, or negative sample replay. However, we are not committed to memory queue data structure, per se. Given that its functional role is to give rise to good representational coverage over the latent space, there may be other architectural mechanisms by which the item separability can be achieved, e.g., enforcing independence across feature channels[86], or including a predictor head[87]. Indeed, there is an ongoing debate about whether the instance-level separability requires these negative samples at all[88–90].

While these instance-level contrastive learning systems advance a domain-general learning algorithm guiding visual representation formation, they are by no means a perfect model of how the brain learns, nor are they a direct model of the biological system per se–we instead see them as a testbed for broader learnability arguments and as useful for providing cognitive insights into visual representation and formats (e.g., clusters in an L2-normed hypersphere can easily be read-out with local linear hyperplanes, and this is not true of euclidean spaces[79]), and as such serve as a useful computational abstraction.

**Concurrent work in non-human primate vision**. In highly related recent work, Zhuang et al., (2021) explored a variety of self-supervised vision models and whether they have brain-like representation, using single-unit responses of the non-human primate ventral visual stream. Broadly, they found that the models using similar instance-level contrastive approaches as ours achieved parity with category-supervised models in predicting single-unit neural response profiles in areas V1, V4, and IT; exceeding the capacities of other kinds of self-supervised models with different goals, including an autoencoder (a reconstructive goal), next frame prediction (PredNet[91]), and other non-relational objectives like depth labeling and colorization[92,93]. Further, they also capitalized on the value of this general objective, developing variations of their instance-level contrastive learning model to learn over video from the SAYcam baby head-cam dataset[60], finding weaker but generally maintained neural predictivity. While almost every methodological detail is different from the work here, including the theoretical assumptions implied by their methods to relate model feature spaces and single-unit responses, these two studies generally drive to very similar broad claims. That is, both provide empirical support for moving away from category-supervision towards instance-level contrastive learning. Further, the differences between our approaches reveal an expansive new empirical space to explore, considering different methods (fMRI, electrophysiology), models (IPCL, Local Aggregation), and model organisms (humans, monkeys); and, critically, the linking hypotheses (veRSA, encoding models) that operationalize our understanding of the neural code of object representation.

**Analytical linking hypotheses between model and brain activations**. The question of how feature spaces learned in deep neural networks should be linked to brain responses measured with fMRI is an ongoing analytical frontier–different methods are abundant[21,22,24,28,94,95], each making different implicit assumptions about the nature of the link between model feature spaces and brain responses. In the present work, we assume a voxel is best understood as a weighted combination of deepnet features— this is intuitive given the coarse sampling of a voxel over the neural population code. However, note that even single neuron responses (measured with electrophysiology in the primate brain) are typically modeled as weighted combinations of deepnet units, or even as weights on the principle components throughout the deepnet feature space[96]. In general, exactly how deepnet units are conceived of (e.g., how the tuning of any one deepnet unit is related to single neuron firing) is still coming into theoretical focus, where different hypotheses are implicit in the kind of regression model (e.g., whether encoding weights should be a sparse and positive relationship, or low in magnitude and distributed across many deepnet units).

To arrive at a single aggregate measure of neural predictivity, encoding model approaches simply average across the set of individual neuron fits[23,65]. In contrast, we considered these voxel-wise encoding models together as an integrated population code,

in which items vary in the similarity of their activation profiles, which focuses on the representational geometry of the embedding[49]. One motivation for this shift to the representational similarity as the critical neural target to predict is that fMRI allows for relatively extensive spatial coverage, providing access to a population-level code at a different scale than is possible with dozens to hundreds of single-unit recordings; indeed trying to predict the RDM of a brain region is now the defacto standard in visual cognitive neuroscience. However, note that our approach differs from other kinds of weighted RSA analyses that are often employed on fMRI data[24,94], which fit the representational geometry directly by re-weighting feature-based RDMs, discarding univariate activation profiles entirely. Finally, for RSA approaches, exactly how distances in high-dimensional feature space are conceived of and computed is a further open frontier[97], where different hypotheses about the way information is evident in the neural code are implicitly embedded in the choice of distance metrics (e.g., as the euclidean distance or the angle between vectors[6,98]).

At stake with these different analytical approaches is that the choices influence the pattern of results and subsequent inferences. For example, in the present data, model features are much more strongly related to brain RDMs when using veRSA than when using classic RSA, which make sense considering this method can recover true relationships that have been blurred by voxel-level sampling; however, untrained models also improve dramatically under this method, raising the question of whether the flexibility of re-weighting to the feature space is too great (or the Pearson-r scoring method is too lenient). As another example, in the present data, the IPCL features were able to comparably capture responses in aOTC to objects at different orientations, but only with veRSA, and not with classic RSA. This discrepancy between the analysis approaches suggests that the brain-like orientation information is embedded in the feature space, but requires voxel-wise encoding models to draw out those relationships—these pairwise relationships are less strongly evident in the unweighted feature space. Why? One possibility is that these IPCL models do not currently experience any orientation jitter across the samples (only crops, resizes, and coloration variation) and thus orientation-tolerance cannot enter into to the instance-prototype representations. In current work, we are adding orientation augmentation to IPCL samples to explore this possibility. More broadly, we highlight these analytic complexities for two reasons. First, to be transparent about the untidy patterns in our data and the current state of our thinking for motivating these analysis decisions in the present work. And second, to open the conversation for the field to understand more deeply the ways in which deepnet models have brain-like representation of visual information under different analysis assumptions, especially as these new interdisciplinary analytical standard approaches are being developed.

**A domain-general account of visual representation learning**. The pressures guiding the tuning along the ventral visual stream, and the formation of object category information, have been deeply debated, with some theories proposing that category-level (or "domain-level") forces may be critical drive the organization of this cortex. That instance-level contrastive learning can result in emergent categorical representation supports an alternative theoretical viewpoint, in which category-specialized learning mechanisms are not necessary to learn representations with categorical structure. On this generalist account, visual mechanisms operate similarly over all kinds of input, and the goal is to learn hierarchical visual features that simply try to discriminate each view from every other view of the world, regardless of the

visual content or domain. We further show that these instance-level contrastive learning systems can have representations that are as brain-like as category-supervised systems, increasing the viability of this general learning account. This generalist view does not deny the importance of abstract categories in higher-level cognition but instead introduces the instance-level learning objective as a proximate goal that learns compact perceptual representations that can support a wide variety of downstream tasks, including but not limited to object recognition and categorization.

## Methods

**Models**. IPCL and category-supervised comparison models were implemented in PyTorch[99], based on the codebase of Wu et al. (https://github.com/zhirongw/lemniscate.pytorch). Code and models are available here: (https://github.com/harvard-visionlab/open_ipcl).

For our primary models, we trained five models with an Alexnet-gn architecture (Supplementary Fig. 1), using instance-prototype contrastive learning (see Supplementary Methods for details), on the ImageNet-1k dataset[100]. We used the data augmentation scheme used by[41], with both spatial augmentation (random crop and resize; horizontal flip), and pixelwise augmentation (random grayscale; random brightness, contrast, saturation, and hue variation). These augmentations require the network to learn a representation that treats images as similar across these transformations. The replications reflect explorations through different training hyper-parameters. See the Supplementary Methods for extended details about the architecture, augmentations, loss function, and training parameters.

For the category-supervised model, we used the same AlexNet-gn architecture as in the primary IPCL models (minus the final L2-norm layer), but with a 1000-dimensional final fully-connected layer corresponding to the 1000 ImageNet classes. The standard cross-entropy loss function was used to train the model on the ImageNet classification task. Otherwise, training was identical to the IPCL models, with the same visual diet (i.e., same batch size and the number of augmented samples per image using the same augmentation scheme), and the same optimization and learning rate settings.

We trained 6 additional IPCL models to examine the impact of visual diet on learned representations, using datasets that focus on objects, places, faces, or a mixture of these image types: (i) ImageNet: ~1.28 million images spanning 1000 object categories[100]. (ii) Objects: OpenImagesV6, ~1.74 million training images spanning 600 boxable object classes (52; 53). (iii) Faces: vggFace2, ~3.14 million training images spanning 8631 face identities[56]. (iv) Places: places2, ~1.80 million images of scenes/places spanning 365 categories[54]; (v) Faces-Places-Objects-1x: a mixture of ImageNet, vggFace, and places2, randomly sampling images across all sets, limited to ~1.28 million images per epoch to match the size of the ImageNet training set, (vi) Faces-Places-Objects-3x: limited to 3.6 million images per epoch. We used less extreme cropping parameters for all of these models than for the primary models so that the faces in the vggFace2 dataset would not be too zoomed in (as in this dataset, they tend to be already tightly cropped views of heads and faces). We used identical normalization statistics for each model (rather than tailoring the normalization statistics to each training set). Finally, we had to reduce the learning rate of the Faces model to .001 in order to stabilize learning. Otherwise, all other training details were identical to those for the primary models.

We also analyzed the representations of several concurrently-developed instance-level contrastive learning models: SimCLR[39]; MoCoV2[38] and SwAV[40], which are trained on ImageNet; and TC-MoCo[59]: trained on baby head-cam video data[60]. These models were downloaded from official public releases.

To extract activations from a model, images were resized to 224 × 224 pixels and then normalized using the same normalization statistics used to train the model. The images were passed through the model, and activations from each model layer were retained for analysis. The activation maps from convolutional layers were flattened over both space and channel dimensions yielding a feature vector with a length equal to NumChannels × Height × Width, while the output of the fully-connected layers provided a flattened feature vector with a length equal to NumChannels.

**fMRI experiments**. The object orientation fMRI dataset reflects brain responses measured in seven participants while viewing images of eight items presented at five different in-plane orientations (0, 45, 90, 135, and 180 degrees), yielding a total of 40 image conditions. These images were presented in a mini-blocked design, wherein each 6 min-12 s run, each image was flashed four times (600 ms on, 400 ms off) in a 4 s block, and was followed by 4 s fixation. All 40 conditions were presented in each run; the order was determined using the optseq2 software and was additionally constrained so that no item appeared in consecutive blocks (e.g., an upright dog, followed by an inverted dog). Two additional 20 s rest periods were distributed throughout the run. Participants completed 12 runs. Their task was to pay attention to each image and complete a vigilance task (press a button when a red circle appeared around an object), which happened 12 times in the run. Participants (ages 20–35, four female, unknown racial distribution) were recruited through the Department of Psychology at Harvard University and gave informed

consent according to procedures approved by the Harvard University Internal Review Board.

The Inanimate Objects fMRI dataset reflects brain responses measured in ten participants while viewing images depicted 72 inanimate items. In each 8-min run, each image was flashed four times (600 ms on, 400 ms off) in a 4 s block, with all 72 images presented in a block in each run (randomly ordered), with $4 \times 15$ s rest periods interleaved throughout. Participants completed six runs. Their task was to pay attention to each image and complete a vigilance task (press a button when a red-frame appeared around an object, which happened 12 times in the run). Participants (ages 19–32; eight females. unknown racial distribution) gave informed consent approved by the Internal Review Board at the University of Trento, Italy.

All fMRI protocols were presented using scripts written in Matlab using PsychToolbox. Functional data were analyzed using Brain Voyager QX software and MATLAB, with standard preprocessing procedures and general linear modeling analyses to estimate voxel-wise responses to each condition at the single-subject level. Details related to the acquisition and preprocessing steps can be found in the Supplementary Information. All analyses were conducted in Matlab and Python using custom analysis code.

*Brain sectors.* First, the EarlyV sector was defined for each individual to include areas V1–V3, which were delineated based on activations from a separate retino-topy protocol. Next, an occipito-temporal cortex mask was drawn by hand on each hemisphere (excluding the EarlyV sector), within which the 1000-most active voxels were included, based on the contrast [all objects > rest] at the group level. To divide this cortex into posterior and anterior OTC sectors, we used an anatomical cutoff (TAL Y: -53), based on a systematic dip in local-regional reliability at this anatomical location, based off on concurrent work also analyzing this Inanimate Object dataset[70]. The same posterior-anterior division was applied to define the sectors and extract data from the Object Orientation dataset.

*Data reliability.* The noise ceiling was defined in each sector, based on splitting participants into two groups and averaging over all possible split-halves. Specifically, we computed all of the subject-specific RDMs for each sector. Then, on a given iteration, we split the participants in half and computed the average sector-level brain RDMs for each of these two groups. We computed the similarity of these two RDMs by correlating the elements along the lower triangular matrix (excluding the diagonal). The correlation distance (1-Pearson) was used for creating and comparing RDMs. This procedure was repeated for all possible split-halves over subjects. The noise ceiling was estimated as the mean correlation across splits (average of fisher-z transformed correlation values), and an adjusted 95% confidence interval that takes into account the non-independence of the samples[101]. This particular method was used to dovetail with the model-brain correlations, described next.

**Model-brain analyses.** The first key dependent measure (veRSA correlation) reflects the suitability of the features learned in a layer to predict the multivariate response structure in that brain sector. To compute this, we used the following procedure.

*Voxel-wise encoding.* For each deepnet layer, subject, and sector, each voxel's response profile (over 40 or 72 image conditions, depending on the dataset) was fit with an encoding model. Specifically, in a leave-one-out procedure, a single image was held out, and ridge regression was used to find the optimal weights for predicting each voxel response to the remaining images. We used sklearn's[102] cross-validated ridge regression to find the optimal lambda parameter. The response for the held-out item was then predicted using the learned regression weights. Each item was held out once, providing a cross-validated estimate of responses to each image in every voxel, which together forms a model-based prediction of neural responses in each brain region. Based on these predicted responses, a model-predicted-RDMs was computed for each participant.

*Layerwise RSA analysis.* Next, for each sector and layer, the model-predicted-RDMs for each subject were divided into two groups and averaged, yielding two average model-predicted-RDMs from two independent halves of the data. Each RDM was correlated with actual brain RDM, where the brain RDM was computed from the same set of participants. This analysis was repeated for all possible splits-halves of the participants. The average fisher-z transformed correlation (and an adjusted 95% confidence interval[101] was taken as the key measure of layer-sector correspondence.

Note that this average correlation reflects the similarity between the model-predicted-RDMs and the brain-RDMs, where only half of the subject's brain data are used. This method of splitting the data into two halves was designed to increase the reliability in the data—we found that the RDMs were more stable with the benefit of averaging across subjects, while any one individual's brain data were generally less reliable. Additionally, this procedure allows there to be some generality across subjects. Finally, we did not adjust the fit values to correct for the fact that the model-to-brain fit reflects only half the brain data, instead, we kept it

as is, which also allows the average layer-sector correlation to be directly compared to the similarly-estimated noise ceiling of the brain data. sector.

*Cross-validated max-layer estimation.* The second key dependent measure relating model-brain correspondence reflects the strength of the best-fitting layer to a given sector. To compute this measure, we again used the same technique of splitting the data in half by two groups of subjects (this time to prevent double-dipping). Specifically, for each model and sector, the veRSA correlation was computed for all layers, and then layer with the highest veRSA correlation was selected. Then, in the independent half of the data (from new participants), the veRSA correlation was computed for this selected layer, and taken as a measure of the highest correspondence between the model and the sector. As above, this procedure was repeated for all possible split-halves of the subjects, and the cross-validated max-r measure was taken as the average across splits (averaging fisher-z transformed correlation values, and using the adjusted 95% confidence interval that takes into account the non-independence of the samples). This procedure insures an independent estimate of the maximum correspondence across layers.

*Classic RSA.* For comparison, we also computed and compared RDMs in both layerwise feature spaces and brain sectors using classic RSA. In this case, RDMs were computed directly from the deepnet activations (across units) and the brain activation patterns (across voxels), with no encoding model or feature weighting.

**Statistical comparisons.** To compare the cross-validated max correlation values between models, we used paired *t*-tests over all split-halves of the data, with a correction for non-independence of the samples, following 101 (tests based on repeated k-fold cross-validation) for corrected variance estimate and adjusted *t*-values. Comparisons between IPCL and Category-Supervised models are found in Supplementary Table 2; Comparisons between IPCL and an untrained model are found in Supplementary Table 2; Comparison between models trained with different visual diets to the baseline IPCL model trained on ImageNet are reported in Supplementary Table 3. Statistical significance for these paired *t*-tests was determined using a Bonferonni corrected $\alpha$ level of $0.05/30 = 0.00167$, where 30 corresponds to the number of family-wise tests for all reported tests.

**Reporting summary.** Further information on research design is available in the Nature Research Reporting Summary linked to this article.

## Data availability
Brain data, analysis code, and figure-plotting code are available on the Open Science Framework (https://osf.io/trne8/). Public image datasets used to train the models include: ImageNet (https://image-net.org/), OpenImagesV6 (https://storage.googleapis.com/openimages/web/index.html), VggFace2 (https://github.com/ox-vgg/vgg_face2), and Places2 (http://places2.csail.mit.edu/). Source data are provided with this paper.

## Code availability
Model training scripts and pretrained models are available on Github (https://github.com/harvard-visionlab/open_ipcl; https://doi.org/10.5281/zenodo.5719364; ref. [103]).

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

## Acknowledgements

Funding for this project was provided by an Amazon Cloud Credits for Research Grant to TK and GAA, an NSF CAREER BCS-1942438 to TK, and an NSF PAC COMP-COG 1946308 to GAA. Thank you to Morgan Henry who collected the Object Orientation dataset.

## Author contributions

Both authors contributed extensively to this work. TK collected and pre-processed the Inanimate Object Dataset. TK and GAA supervised the collection and preprocessing of the Object Orientation dataset. TK organized all brain data for analysis. GAA implemented and trained all models. TK and GA jointly developed the self-supervised model, designed the experiments and analytical procedures, created the figures, and wrote the manuscript.

## Competing interests

The authors declare no competing interests.
