## [Peer Review File · Nature Communications]

Beyond category-supervision: Computational support for domain-general pressures guiding human visual system representationREVIEWER COMMENTS

Reviewer #1 (Remarks to the Author):

Konkle and Alvarez present a thorough series of experiments and analyses to investigate the correspondence between a self-supervised model (based on an instance-prototype contrastive-learning algorithm or IPCL) and fMRI responses in regions of human ventral visual cortex that are thought to be critical for object recognition. They compare this model with the more standard category-supervised models. They find that category information emerges in the self-supervised model and that while the classification performance is weaker than for category-supervised models, the correspondence with the brain is similar suggesting that such self-supervised models provide a potentially viable alternative to the more common supervised models.

The authors are exploring a really interesting direction, but after reading through the whole manuscript I was left a little confused as to what the key take-aways of the work are. Some of the individual analyses are interesting with thought-provoking discussion (e.g. consideration of the untrained models, which I wish more studies would show, and the test of the 'visual diet') but the results themselves are not trivial to interpret. There's probably a whole host of models that can show a similar correspondence with the brain, so how should we choose between them? If the bottom-up experience is so important, why does the nature of the visual diet seem to matter so little? The assumptions built into the models and the training algorithms are critical. In part, I think the manuscript suffers from a little conceptual fogginess in the motivation and nature of the self-supervised model that make me wonder how informative the results really are from a visual neuroscience perspective.

I'll try and elaborate my concerns in the points below:

1) "Prominent theoretical accounts of the organization of the high-level visual system assert that category-level ("domain-level") forces are critical for shaping visual representation"

"Complementing this theoretical perspective, deep convolutional neural network models trained to perform multi-way object categorization learn hierarchical feature spaces that are currently the best predictive models of ventral visual stream responses to object images"

I have two concerns here. First, the authors don't really spell out what they mean by 'category-level' forces. They seem to be contrasting this view with a more feature driven view, but some of the dominant theories of the origin of category-selective regions posit that they arise from the different types of visual features present in, for example, faces versus other objects. So do the authors mean some sort of top-down driven force? Some clarity is really needed here because this is a key part of the motivation for moving to the self-supervised model. Second, 'category' really refers to two different things in these statements. In the brain, the category-selective regions are for groupings of stimuli such as faces, scenes, words or objects in general. In contrast, DNNs are trained on categories at a different level of granularity such as 'balloon', 'strawberry' or even different breeds of dogs. DNNs are not trained with labels corresponding to face, scene or object. So the link between these statements is a little confusing. Again, I think it is important for the authors to provide more clarity here. Ultimately, it strikes me that the authors are using the modeling results to argue for a view of visual cortex and category-selectivity that is not that different from that espoused by, say, Livingstone, Arcaro and colleagues.

2) "...it is clear that humans and non-human primates do not learn visual representation from millions of category labels, and that our perceptual systems discriminate visual objects without requiring category label information."

The motivation for moving from a supervised to a self-supervised model is clear, but given this consideration, it's surprising to me that the authors don't seem too concerned about the plausibility of the self-supervised model. One of the key features of the self-supervised model is the augmented samples. What would these augmented samples correspond to in human visual experience? These augmented samples are somewhat arbitrary and don't seem particularly meaningful. To be fair, the authors are following prior work, but if the goal is to investigate

correspondence with the brain, why not use more meaningful samples? The critical point here is that the model is based around individual images, but human experience is presumably based around individual instances of the same objects. So why not use samples that correspond, for example, to different viewpoints of the same object or different natural lighting conditions or some similar manipulation – crops, rescaling and color jitter are not the same thing. How critical is the nature of the samples?

For me, at least, the results of this manuscript would be much more interesting if the nature of the sampling was a better reflection of human visual experience. I don't know if the results would be similar or not, but I think this is a critical question.

Overall, I think the results the authors present are clear – the self-supervised model can show similar correspondence to the visual cortex as a category-supervised model. But given my concerns about the motivation and nature of the self-supervised model, I'm not sure exactly what I should take away from this. I strongly encourage the authors to provide a more carefully reasoned motivation and to train the self-supervised model in a way that provides a better match to human visual experience. Given these, I think it would be clearer what the implications of the work might be.

Reviewer #2 (Remarks to the Author):

Summary: The authors develop leverage self-supervised contrastive learning to train models of ventral visual cortex. They find that self-supervision yields representational similarity with fMRI recordings that rival conventional supervised learning, and for the most part outperforming random initializations. Since self-supervised learning obviates the need for categorical supervision from human annotators, the authors conclude that it is a superior objective function for modeling neural responses to images.

Strengths: The paper is clearly written and features gorgeous figures. The experiments all make sense, and the logic is clear throughout. I am glad the authors are pushing back on opinions in the field that category-level forces shape the organization of ventral visual areas. This paper joins others (like [1]) indicating that different levels of categorization may not reflect ethological factors.

Weaknesses: The novelty of the paper is an issue. The proposed objective function closely resembles many self-supervised objective functions that have already been published (the authors did a great job reviewing the field). Many of these algorithms perform better on image classification (although as the authors note that may be confounded by model architecture) and fits to electrophys recordings in primate have already been published (Zhuang et al., 2021). The key difference of this paper is that it uses an fMRI dataset. I'm not sure how that helps us understand vision.

Other big issues:

"These performance-based relationships also raise a natural question for the present work—how strongly will models trained with instance-prototype contrastive learning show emergent object category structure, and how well will these features spaces show emergent brain-like representation, relative to their category-supervised counterparts?"

The problem I have with this paper is contrastive learning is no more plausible than the implausible categorical cross-entropy objective function. There's no silver bullet evidence that you need self-supervised learning to (a) perform well on object classification or (b) explain neural data. Rather the evidence is that contrastive learning approaches supervised performance. I'm not sure what this tells us about the brain, other than suggesting that there's other ways of learning

powerful visual representations than contrastive learning. To put a point on it: there's no mechanism in the brain or development where you'd say humans rely on contrastive learning, right? I agree that distinguishing views is plausible, but I'm not sure that plausibility is sufficient for an empirical paper.

"Taken together, these results indicate that the success of CNNs in predicting ventral stream responses is driven by their ability to capture texture-based representations that are also extensively present throughout the ventral stream (Long et al., 2018), but they fall short where more explicit shape representations are emphasized. Capturing brain-like finer-grained distinctions among inanimate objects is thus as an important frontier that is currently beyond the scope of both contrastive and category-supervised models."

I feel like this gap of explainability for aOTC, which might rely on 3D shape-based representations, is the main story in the paper. Identifying an objective function that yields such a representation may not improve object classification on imagenet, but could improve neural explainability beyond the other self-sup and sup methods. That would be an amazing achievement, and I'm disappointed that it's not the focus here.

Less important issues:

"While traditional batch normalization operates by normalizing across images for each feature channel..." You might want to revise this to say "normalize across image batches" whereas groupnorm is individual images. How many groups did you use? One other detail to add is that batchnorm stores the mean/var of feature map activity during training, which is then used at test time.

I'm also curious about what aspect of groupnorm stabilized learning, since you mention its similarity to div norm. Alexnet has LRN, which is an implementation of div norm per-feature column. Group-norm can change the normalization pool size. Is there a sweet-spot for the pool size, and do you have any thoughts on what that means?

Fig1b says MDS vis but the text mentions it's a TSNE.

Fig 1D. What is the unit of the color bar?

You mention "...the group-averaged representational geometries were also highly reliable..." Could you provide more context for this procedure in the same paragraph? The reliabilities are so extremely high that I need at least a hint of methodological detail + reference to later in the paper with the full write-up.

You mention "These findings highlight that there are useful architectural inductive biases present in untrained networks." What's strange about this is that supervision makes an extraordinary difference when modeling image classification, but on this fMRI + RSA analysis it's not so large when comparing the max correlations on the bar graphs in Fig. 2b/c. Could you include the untrained model in the line plot? Perhaps it's just that the max is really high but on average the untrained model is far worse than the others. I think it's essential to have a better explanation for the success of the untrained model than its inductive biases match ventral visual cortex. Or if the inductive biases really are *that* effective, you could rerun the same model, changing its max pool to average pool, or changing its local convs to full-image-sized convs (i.e. MLP), etc.

Is the repulsive term in your objective function critical for explaining neural data? What happens when you have an attraction-only version of your loss, where you just try to make encodings of augmented images as similar as possible. This type of loss resembles an autoencoder and even slow feature analysis. It might help show that your work does something above-and-beyond those classic ML/comp neuro works.

Reviewer #1 (Remarks to the Author):

Konkle and Alvarez present a thorough series of experiments and analyses to investigate the correspondence between a self-supervised model (based on an instance-prototype contrastive-learning algorithm or IPCL) and fMRI responses in regions of human ventral visual cortex that are thought to be critical for object recognition. They compare this model with the more standard category-supervised models. They find that category information emerges in the self-supervised model and that while the classification performance is weaker than for category-supervised models, the correspondence with the brain is similar suggesting that such self-supervised models provide a potentially viable alternative to the more common supervised models.

The authors are exploring a really interesting direction, but after reading through the whole manuscript I was left a little confused as to what the key take-aways of the work are. Some of the individual analyses are interesting with thought-provoking discussion (e.g. consideration of the untrained models, which I wish more studies would show, and the test of the 'visual diet') but the results themselves are not trivial to interpret. There's probably a whole host of models that can show a similar correspondence with the brain, so how should we choose between them? If the bottom-up experience is so important, why does the nature of the visual diet seem to matter so little? The assumptions built into the models and the training algorithms are critical. In part, I think the manuscript suffers from a little conceptual fogginess in the motivation and nature of the self-supervised model that make me wonder how informative the results really are from a visual neuroscience perspective.

I'll try and elaborate my concerns in the points below.

- **To overview our response, your comments helped us get a handle on the points that were confusing conceptually here, related to the category supervisory signal and biological plausibility. For example, to your question: "There's probably a whole host of models that can show a similar correspondence with the brain, so how should we choose between them?" – we now clarify that our aim is not to find the one best model of the brain. Rather, our aim is to make a computational argument about *the kinds of pressures* that are sufficient to arrive at a brain-like representation of object category information; the success of these instance-level models provide a plausibility argument that top-down / external pressures are *not required* to learn brain-like hierarchical visual formats. This work thus provides a theoretical alternative to prominent category-based frameworks of high-level visual representation.**
- **As such, we hope the conceptual fogginess is lifted in the revised introduction, and that the inferential value of the computational existence proof – i.e., showing that category-supervisory signals are not needed to derive a visual feature representation that show comparable brain-like predictivity – is better situated.**

1) "Prominent theoretical accounts of the organization of the high-level visual system assert that category-level ("domain-level") forces are critical for shaping visual representation"

“Complementing this theoretical perspective, deep convolutional neural network models trained to perform multi-way object categorization learn hierarchical feature spaces that are currently the best predictive models of ventral visual stream responses to object images”

I have two concerns here. First, the authors don't really spell out what they mean by 'category-level' forces. They seem to be contrasting this view with a more feature driven view, but some of the dominant theories of the origin of category-selective regions posit that they arise from the different types of visual features present in, for example, faces versus other objects. So do the authors mean some sort of top-down driven force? Some clarity is really needed here because this is a key part of the motivation for moving to the self-supervised model. Second, 'category' really refers to two different things in these statements. In the brain, the category-selective regions are for groupings of stimuli such as faces, scenes, words or objects in general. In contrast, DNNs are trained on categories at a different level of granularity such as 'balloon', 'strawberry' or even different breeds of dogs. DNNs are not trained with labels corresponding to face, scene or object. So the link between these statements is a little confusing. Again, I think it is important for the authors to provide more clarity here. Ultimately, it strikes me that the authors are using the modeling results to argue for a view of visual cortex and category-selectivity that is not that different from that espoused by, say, Livingstone, Arcaro and colleagues.

- **Thanks for these very on-point comments. We have re-written the introduction (1) to clarify what we mean by “category-level pressures”, and (2) to highlight the discrepancy between broader category distinctions typically posited in cognitive neuroscience theories and the finer-grained supervision of category-trained deep neural networks. From our perspective, having to think about these issues more carefully has led to a much stronger and theoretically clearer introduction about what we are testing, and what we learning about the visual system, from this work.**
- **Further, you are correct; we are indeed espousing a view that is deeply compatible with that of Arcaro and Livingstone, critically providing new coupled empirical and computational evidence to this end; we now highlight this link in the revised manuscript.**

2) “...it is clear that humans and non-human primates do not learn visual representation from millions of category labels, and that our perceptual systems discriminate visual objects without requiring category label information.”

The motivation for moving from a supervised to a self-supervised model is clear, but given this consideration, it's surprising to me that the authors don't seem too concerned about the plausibility of the self-supervised model. One of the key features of the self-supervised model is the augmented samples. What would these augmented samples correspond to in human visual experience? These augmented samples are somewhat arbitrary and don't seem particularly meaningful. To be fair, the authors are following prior work, but if the goal is to investigate correspondence with the brain, why not use more meaningful samples?

- **We have actually removed the motivation of getting rid of labels--and the even word “biological plausibility”--entirely from the revised manuscript (!). Thanks to your earlier comments, we now more clearly understand our motivation, which is not**

about making a better mechanistic model of the biological system; but rather, to understand the kinds of representational goals and pressures that underlie useful visual representation formation. That is, we are using these models primarily as a framework for learning visual representation, at a more cognitive level of analysis. And, we use links between these features spaces and the structure of brain responses to highlight that these kinds of representations are compatible.

- Of course, we think there are really interesting links between the way the model learns and how humans might learn (at a rather coarse level of abstraction). In particular, the model takes samples from the world, like an active learner: this creates a framework in which the model itself must take ‘an action’ to learn good visual structures (c.f. Held and Hein’s cat studies). We are currently thinking more deeply about what kind of action policies, etc, will be key for learning good structure in a more active-sensing framework. Will it be critical to have a foveated sensor, with different kinds of visual image information linked between samples? Or a predictive error signal in the latent space between successive samples? So, these are open questions which we are currently thinking about in order to make the samples ‘more meaningful.’ We hope the revised framing of the manuscript will allow the reader to see these open questions as exciting frontiers, rather than problems with the model.

The critical point here is that the model is based around individual images, but human experience is presumably based around individual instances of the same objects. So why not use samples that correspond, for example, to different viewpoints of the same object or different natural lighting conditions or some similar manipulation – crops, rescaling and color jitter are not the same thing. How critical is the nature of the samples?

- The directions we are taking next are about considering this model as an active observer; so, consistent with what you say, we will move away from ‘picture plane’ augmentations like crops, rescaling, and jitter, and towards the kinds of structure variation that is available when you move around the world. There does not yet exist a large scale dataset of natural images that can be used for this purpose (e.g., Facebook’s recently released CO3D has only 50 object categories), and we have not found a synthetic dataset that has the scale and quality needed, though we’re hopeful that recent advances in this area (e.g., Facebook’s Habitat2 environment) will enable this work to be done in the future.
- To answer your broader question: as the field of self-supervised learning is advancing the question of ‘how critical is the nature of the samples’ is increasingly being explored, and the answers to date seem to be that they are *very* important. For example, in a systematic ablation study, Chen et al. (2020) found that no single augmentation suffices to learn good representations, and that the combination of random cropping and random color distortion leads to particularly useful visual representations with their contrastive learning algorithm (SimCLR). Some have argued that contrastive samples should reduce the mutual information (MI) between views while keeping task-relevant information intact (Tian et al., 2020), and the combination of spatial augmentation with color distortion seems to fit these requirements well. This suggests it is important for future research to understand computationally why color distortion is so useful when combined with spatial augmentation, and whether the benefits/constraints of color distortion can be

approximated by naturalistic sampling in the way that spatial augmentations can be. Alternatively, it may be important to separate out the analysis of spatial and color information (which occurs in primate vision), essentially treating these as pathways as distinct "views" of the scene (e.g., see contrastive multi-view coding for an example of how these can be handled within a CL framework, Tian et al., 2019). In sum, the augmentations are critical, and a theory of why and what and how is still emerging.

For me, at least, the results of this manuscript would be much more interesting if the nature of the sampling was a better reflection of human visual experience. I don't know if the results would be similar or not, but I think this is a critical question.

- **We agree that capturing samples that reflect visual experience is truly an exciting frontier. And, we humbly point out that while capturing human visual experience is the longer-term goal, there are many steps to work out en route. This is more like a multi-step, research program (e.g. adding a foveal transform to the model, considering different sampling policies, operating in a 3-dimensional world with lighting and viewpoint changes, rather than picture plane statistics).**
- **Our model came out at the same time as the self-supervised models in the computer vision community, developed concurrently rather than in reference to them. And, the work reported in this manuscript is the first to relate self-supervised visual feature spaces to the response structure of the human brain, in two condition-rich fMRI datasets, developing new methods for linking these between models and brains. Finally, this work provides a new kind of evidence advocating for the plausibility of a domain-general account of visual representation learning. We hope you concur that these reflect important empirical-computational advances; even though there is always more work to do.**

Overall, I think the results the authors present are clear – the self-supervised model can show similar correspondence to the visual cortex as a category-supervised model. But given my concerns about the motivation and nature of the self-supervised model, I'm not sure exactly what I should take away from this. I strongly encourage the authors to provide a more carefully reasoned motivation and to train the self-supervised model in a way that provides a better match to human visual experience. Given these, I think it would be clearer what the implications of the work might be.

- **Thank you for your review. The delay in our reply is in part because we really did try to follow your points and have begun to train models in various ways that might "better match human visual experience" but it is a really wide landscape (and each model takes 10 days to train and we can train 3 at a time; we have even worked to find new compute infrastructure to reduce the cost of training these models and increase the speed). In starting this work, it has become clear to us that capturing the structure of human visual experience is really the scope of another (if not several!) papers.**
- **In response, we have attempted to clarify the theoretical importance of demonstrating that these kinds of self-supervised models have comparable brain**

predictivity—not as better models of the brain per se, but rather as serving at the level of abstraction of a learnability argument: this work provides an actual proof of concept that category-level information can arise without category-level supervisory pressures, with visual features that are comparable predictors of the structure in brain responses to objects.

Reviewer #2 (Remarks to the Author):

Summary: The authors develop leverage self-supervised contrastive learning to train models of ventral visual cortex. They find that self-supervision yields representational similarity with fMRI recordings that rival conventional supervised learning, and for the most part outperforming random initializations. Since self-supervised learning obviates the need for categorical supervision from human annotators, the authors conclude that it is a superior objective function for modeling neural responses to images.

Strengths: The paper is clearly written and features gorgeous figures. The experiments all make sense, and the logic is clear throughout. I am glad the authors are pushing back on opinions in the field that category-level forces shape the organization of ventral visual areas. This paper joins others (like [1]) indicating that different levels of categorization may not reflect ethological factors.

- **Thanks!**

Weaknesses: The novelty of the paper is an issue. The proposed objective function closely resembles many self-supervised objective functions that have already been published (the authors did a great job reviewing the field). Many of these algorithms perform better on image classification (although as the authors note that may be confounded by model architecture) and fits to electrophys recordings in primate have already been published (Zhuang et al., 2021). The key difference of this paper is that it uses an fMRI dataset. I'm not sure how that helps us understand vision.

- **We believe there are several aspects of our work which make it distinct from Zhuang et al., both empirically and in terms of the theoretical emphasis and conceptual contribution.**
- **In particular, we believe that probing population-level response structure in human fMRI with different datasets is actually importantly not the same thing as single-unit predictivity of responses to object images in macaque IT — e.g., we cannot know whether results that hold for single units in monkey brains will hold for the large-scale population structure of human cortex (particularly when considering the role of categories on representation learning!).**
- **Further, the theoretical arguments Zhuang et al. make in their paper are actually rather different than ours, as they have a more neuroscience model building aim (i.e., to model biology), and ours is focused more on cognitive architecture and representation learning (abstracted from artificial and biological implementation using representational similarity analysis). As such, we think our work is**

complementary with theirs, and that the revised manuscript makes our unique contribution more clear (focusing less on biological plausibility and more on cognitive architecture and representation learning algorithms; also thanks to Reviewer #1 on this point).

- **Additionally, in truth, much of this work you highlight that detracts from our novelty was done concurrently with the present work. For example, the Zhuang et al., 2021 paper was initially published as a preprint on June 18th, 2020 within 2 days of ours which was on June 16th, 2020. Indeed, at this time just over year ago, even the first major self-supervised model successes like SimCLR and MoCo models had only just been put up as preprints a few months before our initial preprints in Feb 2020 and March 2020. We included these models for comparison with ours, but they did not inform our model development. So the combination of a slower timeline in cognitive neuroscience (vs CS), in addition to the effects of the pandemic on work hours (we had kids age 3 and 5 at the time), meant the pace of our manuscript revising was slower than Zhuang et al.**
- **In sum, we hope that considering not only the similar initial timing of the preprints, but also the more critical factors related to the empirical differences in the key dependent measures, the auxiliary explorations, and the theoretical scope and claims of the introduction and discussion, helps support the case for both the novelty and timeliness of our work.**

Other big issues:

"These performance-based relationships also raise a natural question for the present work—how strongly will models trained with instance-prototype contrastive learning show emergent object category structure, and how well will these features spaces show emergent brain-like representation, relative to their category-supervised counterparts?"

The problem I have with this paper is contrastive learning is no more plausible than the implausible categorical cross-entropy objective function. There's no silver bullet evidence that you need self-supervised learning to (a) perform well on object classification or (b) explain neural data. Rather the evidence is that contrastive learning approaches supervised performance. I'm not sure what this tells us about the brain, other than suggesting that there's other ways of learning powerful visual representations than contrastive learning.

- **We believe our contribution is not to propose a biological model, but instead a cognitive framework (self-supervised visual representation learning without top-down category knowledge), which challenges the prevailing view in visual cognitive neuroscience that category pressures are essential for shaping ventral stream representations (which has prevailed in large part due to a lack of a viable alternative, which we now provide). Before this kind of work, empiricists could only make just-so stories that category information was probably available purely from compressions of the input; while nativists would argue that surely there are so many ways the input could be represented that some supervisory signal would be critical to learn the right kinds of statistics (and these debates still wage on!). But the present work, and instance level self-supervised learning more generally, provides for the first time a way to think about how a visual system might have emergent “high-level” information (present in population level representational structure) without presupposing high-level mechanisms are involved in shaping it.**

- **To clarify this argument, in the revised manuscript, we have removed the somewhat-superficial motivation that these models are ‘biological plausible’, and instead state that our inferential purchase comes from treating them as abstractions that enable us to ask broad questions about the impact of the input and task on representation learning, focusing on the representational signatures of both deepnets and brains, which explicitly abstracts beyond the implementation details (both artificial and biological).**

To put a point on it: there's no mechanism in the brain or development where you'd say humans rely on contrastive learning, right? I agree that distinguishing views is plausible, but I'm not sure that plausibility is sufficient for an empirical paper.

- **While secondary to our main claims, we do think it is worthwhile to consider whether and how both category-supervised and contrastive-learning models could be instantiated in the brain. This is where we have to walk a careful line (and thanks to the reviewer for pointing out issues with our initial framing, which helped us clarify this for ourselves). Although these models aren't intended as models of biology, we can discuss how their computations may be mechanistically grounded. In the case of category-supervision, the idea is that high-level categorical/semantic information impacts visual representation learning through top-down feedback mechanisms via whole-brain networks (c.f. the argument made in Mahon & Caramazza, 2013). In the case of contrastive-learning, the mechanisms that have been posited in the “predictive coding” theory of Rao and Ballard (where discrepancies between predicted and actual sensory states trigger learning) can be adapted to contrastive learning over visual samples. In the revised manuscript we have attempted to separate the main computational learnability argument from the subsequent implications for specific biological mechanisms, saving the latter for general discussion.**

"Taken together, these results indicate that the success of CNNs in predicting ventral stream responses is driven by their ability to capture texture-based representations that are also extensively present throughout the ventral stream (Long et al., 2018), but they fall short where more explicit shape representations are emphasized. Capturing brain-like finer-grained distinctions among inanimate objects is thus as an important frontier that is currently beyond the scope of both contrastive and category-supervised models."

I feel like this gap of explainability for aOTC, which might rely on 3D shape-based representations, is the main story in the paper. Identifying an objective function that yields such a representation may not improve object classification on imagenet, but could improve neural explainability beyond the other self-sup and sup methods. That would be an amazing achievement, and I'm disappointed that it's not the focus here.

- **We agree that the gap of explaining aOTC is important (and also differs from Zhuang et al's conclusions!). 3D shape is one possible account of the differences (with some supporting evidence for this view in forthcoming brain-behavioral work from our lab), but there are also other potential features (e.g. similarity in the Motor plans evoked to interact with these objects). It is also the case that in addition to**

changing the goal, the architecture may be key (e.g. we are examining vision transformer models now).

- **Thus to clarify, the aim of the present work is to make a case for a domain-general learning rule underlying high-level visual representation, which we think is theoretically important in its own right, given the current dominance of category-specialized learning mechanisms underlying the formation of high-level visual responses. In doing so, we also discovered a brain predictivity gap that *both* category-supervised and self-supervised models fail to achieve—in ongoing work we are tackling this new puzzle to understand whether architecture, task, or some combination are critical for closing this gap. We take the reviewer’s enthusiasm for this particular aspect of the paper as a positive, indicating that the present findings pose a significant challenge which will drive future work that is of broad interest.**

Less important issues:

"While traditional batch normalization operates by normalizing across images for each feature channel..." You might want to revise this to say "normalize across image batches" whereas groupnorm is individual images. How many groups did you use? One other detail to add is that batchnorm stores the mean/var of feature map activity during training, which is then used at test time.

- **We have made these changes. We used 32 groups per layer, so as the number of channels per layer increased with depth, the number of channels per group increased with depth (3, 8, 12, 12 for conv2-5).**

I'm also curious about what aspect of groupnorm stabilized learning, since you mention its similarity to div norm. Alexnet has LRN, which is an implementation of div norm per-feature column. Group-norm can change the normalization pool size. Is there a sweet-spot for the pool size, and do you have any thoughts on what that means?

- **We haven't explored what aspect of group norm stabilized learning. To some extent we are compute-limited, as training these models can take ~10 days (+2 for analysis). Given that our IPCL models achieved parity with category-supervised models, we focused our resources on training models that we thought would provide further insight into the comparison between category-supervised and IPCL models (e.g., the visual diet variations).**

Fig1b says MDS vis but the text mentions it's a TSNE.

- **Changed, it is in fact TSNE.**

Fig 1D. What is the unit of the color bar?

- **The unit was Pearson-r. We have added this information to the figure and caption.**

You mention "...the group-averaged representational geometries were also highly reliable..." Could you provide more context for this procedure in the same paragraph? The reliabilities are so extremely high that I need at least a hint of methodological detail + reference to later in the paper with the full write-up.

- **We have made this change in the revision, emphasizing that we used an fMRI protocol that provides reliable estimates of responses to individual items (Tarhan & Konkle, 2020), and that the resulting individual representational geometries were reliable and consistent across subjects, yielding highly reliable group representational geometries.**

You mention "These findings highlight that there are useful architectural inductive biases present in untrained networks." What's strange about this is that supervision makes an extraordinary difference when modeling image classification, but on this fMRI + RSA analysis it's not so large when comparing the max correlations on the bar graphs in Fig. 2b/c. Could you include the untrained model in the line plot? Perhaps it's just that the max is really high but on average the untrained model is far worse than the others. I think it's essential to have a better explanation for the success of the untrained model than its inductive biases match ventral visual cortex. Or if the inductive biases really are *that* effective, you could rerun the same model, changing its max pool to average pool, or changing its local convs to full-image-sized convs (i.e. MLP), etc.

- **In the revised manuscript, we now address the counterintuitive finding that untrained models are doing better than expected: "A deeper inspection revealed that the predicted representational distances in untrained models hover around zero, which is consistent with the fact that they cannot classify object categories very well. However, these features spaces nevertheless contain small differences that are consistent with the brain data, amplified by the voxel-wise encoding procedure."**
- **We have not added the untrained model to the line plot because it obscures the comparison between category-supervised and IPCL models (which is our primary focus), but do reference the layerwise fits which are shown in Supplementary Figure 2. To answer your question, this figure reveals that the untrained models' correspondence with brain responses appears to peak in early model layers and then is sustained deeper into the network. So, interestingly, it isn't just that the max is really high, but instead there appears to be some information that accumulates over the first few layers and is then maintained into later layers.**
- **We also explored a few untrained model variations, varying kernel sizes from .5x – 16x (relative to AlexnetGN baseline, conv1 and conv2 only), using both MaxPool and AvgPool layers. To summarize briefly, we find similar results for kernel sizes from .5x-2x, but beyond that correspondence with brain responses falls off as kernel size increases (for all brain regions in both datasets, for veRSA and classic RSA). Thus, the fits of untrained models to early visual responses appears to reflect hierarchical extraction of relatively localized structure. We have not included these analyses in the paper, because they are tangential to the primary point we are trying to make, but we can if the reviewer feels they are necessary.**

Is the repulsive term in your objective function critical for explaining neural data? What happens when you have an attraction-only version of your loss, where you just try to make encodings of augmented images as similar as possible. This type of loss resembles an autoencoder and even slow feature analysis. It might help show that your work does something above-and-beyond those classic ML/comp neuro works.

- **We think these are really interesting questions that we hope will be explored in follow up research, but which are orthogonal to the claims of the current paper (which focus on parity between supervised and contrastive learning models). Recent models that have only attractive terms (SimSiam and BarlowTwins) use various other tricks to stabilize the training, and we suspect that such modifications would be needed if we removed the repulsive term (and may indirectly achieve the same goal of separability amongst representations). While we think this additional modeling work is beyond the scope of the present paper, we now cite these attraction-only self-supervised models in the General Discussion when addressing possible role of the contrastive term implemented by our memory queue.**

REVIEWERS' COMMENTS

Reviewer #1 (Remarks to the Author):

I have to say I have really enjoyed this discussion with the authors (albeit through the stilted medium of peer review!). The original manuscript provoked many thoughts and the authors have done a fantastic job of grappling with the concerns I raised. From my viewpoint, the revised version of the manuscript is much stronger because the theoretical framing is much clearer. I think this is a very pertinent and timely manuscript and I have no further issues to raise. I look forward to seeing the discussion this provokes in the community!

Reviewer #2 (Remarks to the Author):

"In particular, we believe that probing population-level response structure in human fMRI."

- Speaking as someone with a background in fMRI, this seems like it would be an indictment of the imaging method rather than the modeling if the result was negative. That said, this is a good point to raise about the distinction of the current work with Zhuang et al.

"...the theoretical arguments Zhuang et al. make in their paper are ... neuroscience model building..., and ours is focused on cognitive architecture and representation learning."

- I'm not sure if I buy this as a motivation for the current work. If you were interested in cognitive architecture and representation learning why use fMRI at all? Why not just look at behavior? I clearly don't totally understand or appreciate the point you are trying to make, so I hope you can clarify a bit more. This feels handwavy.

"Additionally, in truth..."

- I definitely appreciate the struggle these past two years have been, particularly with balancing science and child care. I take back my initial critique and I am not going to penalize you for coming second *in publication but basically not in preprint* in the horse race w/ Zhuang et al.

- I'm glad you revised the biological plausibility claims.

- I think the additional assay of untrained models helps unpack that result from the main text. I don't think you need to go into aching detail, but adding the blurb you have in this response to the SI might satisfy inquiring minds, even if the point is somewhat tangential.

Overall, I'm satisfied with these responses and recommend this paper for publication.

REVIEWERS' COMMENTS

Reviewer #1 (Remarks to the Author):

I have to say I have really enjoyed this discussion with the authors (albeit through the stilted medium of peer review!). The original manuscript provoked many thoughts and the authors have done a fantastic job of grappling with the concerns I raised. From my viewpoint, the revised version of the manuscript is much stronger because the theoretical framing is much clearer. I think this is a very pertinent and timely manuscript and I have no further issues to raise. I look forward to seeing the discussion this provokes in the community!

- **We feel exactly the same way! Thank you for helping us make this a better manuscript.**

Reviewer #2 (Remarks to the Author):

"In particular, we believe that probing population-level response structure in human fMRI [with different datasets is actually importantly not the same thing as single-unit predictivity of responses to object images in macaque...]."

- Speaking as someone with a background in fMRI, this seems like it would be an indictment of the imaging method rather than the modeling if the result was negative. That said, this is a good point to raise about the distinction of the current work with Zhuang et al.

- **We are glad you agree this is a good point to raise.**

"...the theoretical arguments Zhuang et al. make in their paper are ... neuroscience model building..., and ours is focused on cognitive architecture and representation learning."

- I'm not sure if I buy this as a motivation for the current work. If you were interested in cognitive architecture and representation learning why use fMRI at all? Why not just look at behavior? I clearly don't totally understand or appreciate the point you are trying to make, so I hope you can clarify a bit more. This feels handwavy.

- **We think that human brain response structure and organization give important insights into the nature of the visual representational format, which in turn provides one source of insight into cognitive architecture—a source of convergent information that complements but is different than what one can measure with behavior (c.f. Konkle & Caramazza, JNeuro, 2013). Given your comment, and other conversations happening around what it means to 'understand' visual representation, we are working on an opinion paper directly aimed at clarifying this broad argument.**

"Additionally, in truth [much of this work you highlight that detracts from our novelty was done concurrently with the present work...]"

- I definitely appreciate the struggle these past two years have been, particularly with balancing science and child care. I take back my initial critique and I am not going to penalize you for

coming second *in publication but basically not in preprint* in the horse race w/ Zhuang et al.

- **This is quite a relief.**

- I'm glad you revised the biological plausibility claims.

- **Thanks.**

- I think the additional assay of untrained models helps unpack that result from the main text. I don't think you need to go into aching detail, but adding the blurb you have in this response to the SI might satisfy inquiring minds, even if the point is somewhat tangential.

- **We have added this text into the SI.**

Overall, I'm satisfied with these responses and recommend this paper for publication.

- **Hurray! Thanks for your time and attention spent reviewing this work.**